# Antiangiogenic Therapy in Clear Cell Renal Carcinoma (CCRC): Pharmacological Basis and Clinical Results

**DOI:** 10.3390/cancers13235896

**Published:** 2021-11-24

**Authors:** Alessandro Comandone, Federica Vana, Tiziana Comandone, Marcello Tucci

**Affiliations:** 1Department of Oncology, San Giovanni Bosco Hospital, 10100 Torino, Italy; alessandro.comandone@aslcittaditorino.it; 2Italian Group of Rare Tumors, Corso Galileo Ferraris 54, 10129 Torino, Italy; tiziana.comandone@edu.unito.it; 3School of Hospital Pharmacy, University of Torino, 10100 Torino, Italy; 4Department of Oncology, Cardinal Massaia Hospital, 14100 Asti, Italy; mtucci@asl.at.it

**Keywords:** renal cancer, angiogenesis, tyrosine kinase inhibitors, immune checkpoint inhibitors

## Abstract

**Simple Summary:**

In the last 15 years, a deep improvement in the knowledge regarding the biological mechanisms responsible for neoplastic cell development and progression has led to a dramatic change in the treatment landscape of metastatic clear cell renal carcinoma. Nowadays, it is known that neo-angiogenesis is a key player in tumor growth and metastatic spread. In particular, the crucial role of the mutation of the von Hippel–Lindau (VHL) tumor suppressor gene, leading to angiogenesis through the transcription of multiple pro-angiogenic factors, is clearly recognized. On the basis of this biological evidence, three classes of targeted therapies with antiangiogenetic activity have received approval for the treatment of advanced disease: tyrosine kinase inhibitors (TKIs); a monoclonal antibody that interferes with vascular endothelial growth factor (VEGF); and two mammalian target of rapamycin (mTOR) inhibitors. These drugs showed impressive results in terms of progression-free survival and objective response rate. In addition, a “second therapeutic revolution” has recently started, due to the latest information on the immunogenic characteristics of renal cell carcinoma and the interplay between angiogenesis and immune surveillance systems. Consequently, immune checkpoint inhibitors, alone or in combination with TKIs, have been approved. In this review, we analyze the pharmacological characteristics and activity of antiangiogenic drugs approved for the treatment of metastatic clear cell renal carcinoma.

**Abstract:**

Angiogenesis has a direct stimulatory effect on tumor growth, duplication, invasion and metastatic development. A significant portion of conventional renal cell carcinomas are angiogenesis-dependent tumors and the pathways supporting this process have been thoroughly investigated over the last 20 years. As a consequence, many tyrosine kinase inhibitors (TKIs) (sunitinib, sorafenib, pazopanib, axitinib, and cabozantinib), one monoclonal antibody (bevacizumab), and two mammalian target of rapamycin (mTOR) inhibitors (temsirolimus and everolimus) have been investigated and approved for the treatment of advanced or metastatic clear cell renal carcinoma (metastatic CCRC) in first-line, as well as second-line, therapy, with impressive results in progression-free survival and in the objective response rate compared with previously available therapies or placebo. Recently, a new type of drug has been approved for metastatic CCRC: immunomodulatory checkpoint inhibitors (ICIs), alone or in combination with TKIs. However, many questions and areas to be explored still remain with regard to clear cell renal carcinoma (CCRC) treatment: research on predictive biomarkers, the best patient selection, how to overcome the mechanisms of resistance, and the best sequence of therapies in daily clinical practice. This review focuses on the pharmacological properties and anticancer activities of these drugs. The toxicity profile and clinical limitations of these therapies are also discussed.

## 1. Introduction

Kidney cancer represents the 7th most common cancer in men and the 10th most common cancer in women [1]. It has been found that 30% of patients have metastatic disease ab initio and almost 30% of the remaining patients will develop metastases. Only 12% of metastatic patients are alive after 5 years [1]. Clear cell renal carcinoma (CCRC) accounts for nearly 80% of all kidney cancers [1]. Other histopathological types are papillary renal cell carcinoma (10–16%) and chromophobe renal cell carcinoma (5%) [1].

CCRC is known to be a heterogeneous disease characterized by different genetic aberrations (mutation, deletion, or hypermethylation) [2,3]. The most common mutation is found on the von Hippel–Lindau (VHL) gene, a tumor suppressor gene that has a key role in angiogenesis [4,5]. Angiogenesis plays a pivotal role in the growth and development of CCRC and can be a target of vascular endothelial growth factor receptor (VEGFR) inhibitors, tyrosine kinase inhibitors (TKIs), and immunotherapy [6,7,8]. The therapeutic armamentarium has evolved over time. Before the introduction of the more recent immunologic therapies for metastatic clear cell renal carcinoma (metastatic CCRC), antiangiogenic and anti-mammalian target of rapamycin (mTOR) agents represented the cornerstone of treatment for over 10 years [9,10]. The introduction of immunotherapy provided a dramatic change in the treatment landscape of metastatic CCRC. In November 2015, the US Food and Drug Administration (FDA) approved nivolumab, a checkpoint inhibitor, for the treatment of metastatic CCRC patients progressed after treatment with antiangiogenic agents on the basis of the CheckMate-025 study. This trial showed significant activity in terms of the overall survival (OS) of nivolumab compared with everolimus in 821 metastatic CRCC patients who had received prior antiangiogenic therapy (25 months versus 19.6 months; HR: 0.73 (98.5% CI, 0.57 to 0.93); *p*= 0.002) [8]. In addition, recent results obtained with immunotherapy combinations have marked the start of the “golden age” of metastatic CCRC treatment [11].

Despite this progress, few active biomarkers are useful in metastatic CCRC as prognostic as well as predictive factors in daily clinical practice [12,13]. Nowadays, the most remarkable negative prognostic markers are identified in the Memorial Sloan–Kettering Cancer Center (MSKCC) risk index: Karnofsky performance status <80%, high serum lactate dehydrogenase (LDH) and calcium level, low hemoglobin, and short interval from diagnosis of primary tumor to appearance of metastatic disease (Table 1) [14]. In the International Metastatic RCC Database Consortium (IMDC) risk score, high neutrophil and platelet counts are considered two other negative prognostic factors (Table 2) [15].

In our review, we discuss the biological role of angiogenesis and the crucial interplay between angiogenesis and the immune system in CCRC pathogenesis. In addition, we analyze the pharmacological characteristics and activity of the antiangiogenic drugs approved for the treatment of metastatic CCRC, alone or in combination with immunological agents.

## 2. Angiogenesis: The Key Player in Development and Progression of Kidney Cancer

Angiogenesis is a physiological process required to generate new blood vessels [16,17]. Neo-angiogenesis is strategic in wound healing, embryo development, and organ perfusion [16,17], as well as in cancer growth, invasion, and metastatic development [16,17]. The balance between angiogenesis-inducing factors and angiogenesis inhibitor pathways is crucial in the development process [17]. Oxygen tension is the most important regulator. Endothelial and smooth muscle cells interact with the three isoforms of hypoxia-inducible factor (HIF): HIF 1-2-3. The interaction causes the synthesis of many proteins responsible for angiogenesis regulation [9]. VEGF, epidermal growth factor (EGF), fibroblast growth factor (FGF), prostaglandin, and tumor necrosis factor-α (TNF)-α stimulate neo-angiogenesis [2,6], while angiostatin, thrombospondin 1 and 2, angiopoietin, endostatin, osteospontin, vasostatin, and cellular communication network factor 3 (CCN3) act as direct angiogenesis inhibitors [17]. These physiological mechanisms also characterize cancer cells [17].

Angiogenesis inhibitors target tumor and stromal cells hampering the expression of pro-angiogenic factors such as EGF, VEGF, and their receptors (EGFR, VEGFR), or interfering with the mammalian target of rapamycin (mTOR) signaling pathway. VEGF activity is mediated through binding to its receptor (VEGFR 1-2-3). This receptor forms a complex with an intracellular tyrosine or serine/threonine kinase. The activated receptors lead to the production of multiple second messengers inducing specific gene expression and thereby modulating the proliferation, survival, migration, and permeability of vascular endothelial cells [18]. mTOR is the pivotal component of the phosphatidylinositol 3-kinase/protein kinase B (PI3K/AKT) signaling pathway that regulates many biological activities, including protein synthesis, angiogenesis, and autophagy. mTOR signaling deregulation is also implicated in cancer progression [12,15].

CCRC is frequently characterized by the inactivation or mutation of the VHL gene, a tumor suppressor gene, leading to angiogenesis through the transcription of potent pro-angiogenic factors regulated by HIF, such as VEGF [13]. The VHL gene encodes for the VHL protein, which interferes with the subunit HIF-1α. In the case of VHL gene alteration, an accumulation of HIF-1α occurs. Increased HIF-1α levels induce the hypoxic response of cells, causing the transcription of multiple oncogenic genes responsible for pro-angiogenic factor production and cell proliferation. These biological processes can eventually induce the development of highly vascularized tumors [19,20]. 

## 3. Antiangiogenic Drugs: Pharmacological Characteristics and Clinical Activity

### 3.1. Monoclonal Antibodies

Bevacizumab is a recombinant humanized anti-VEGF monoclonal antibody [21] binding the circulatory VEGF and preventing the activation of VEGFR on the endothelial cell surface [21].

In metastatic CCRC, as single agent, bevacizumab showed superior activity compared with the placebo [22]. The agent was approved for use in a metastatic setting in combination with interferon alpha (INFα) after the publication of two different studies, in which bevacizumab was administered at 10 mg/kg at 15-day intervals and INFα 9 MU three times a week [23,24,25]. The phase III, randomized, double-blinded AVOREN study led to an overall response rate (ORR) of 31% for the combination versus 13% for INF alone; progression-free survival (PFS) was 10.2 versus 5.4 months [23]. No differences in OS were observed [23]. The most common side effects of the combination arm were hypertension, proteinuria, embolism, wound bleeding, and asthenia [23]. The CALGB study showed an ORR of 25% for the combination of bevacizumab plus INFα versus 13% for INFα alone. The PFS was 8.5 months versus 5.2 months in favor of the combination. No difference in OS was seen [24,25].

According to the European Society of Medical Oncology (ESMO) guidelines, the combination is approved for use in metastatic CCRC treatment with good or intermediate prognosis till progression or toxicity.

### 3.2. Small Molecules and Tyrosine Kinase Inhibitors (TKIs)

Tyrosine kinases are enzymes that catalyse the transfer of a phosphate group from adenosine triphosphate (ATP) to a specific amino acid component of cytoplasmic proteins [26,27,28]. Tyrosine kinases include receptor tyrosine kinases and non-receptor tyrosine kinases [29]. The receptors are transmembrane proteins activated by a physiological ligand [29]. Ligand binding induces the dimerization of TK receptors and the consequent phosphorylation of the cytoplasmic domains [29,30]. The final result is the activation of crucial intracellular pathways responsible for cell duplication. Nowadays, we recognize more than 60 TK receptors [29]. Receptor auto-phosphorylation is a frequent event in cancer cells with the activation of neo-angiogenesis pathways [30]. 

TKIs are synthetic drugs characterized by different sites and amplitudes of activity, and since the beginning of 21st century, they have represented the prototype of targeted therapy [31]. Angiogenesis is one of the most important targets for TKI therapy, and metastatic CCRC represents an extraordinary example of the significant activity of this therapy.

#### 3.2.1. Sunitinib

Sunitinib (SU11248) was the first molecule to radically modify the treatment landscape of metastatic CCRC. It is a small molecule that exerts antiangiogenic activity interfering with multiple targets: VEGFR1, 2 and 3; c-kit; platelet-derived growth factor receptor (PDGFR); FMS-related receptor tyrosine kinase 3 (FLT3); colony stimulating factor 1 receptor (CSFR1); and neurotrophic factor. These receptors are involved in tumor growth, neoangiogenesis and metastatic progression [32,33].

Sunitinib was investigated in two phase II trials of metastatic CCRC cytokine-refractory patients, showing a high rate of objective responses (40% and 34%, respectively) and an unprecedentedly long time to progression (TTP) of 8.7 months [34,35]. The pivotal phase III trial randomized 750 untreated metastatic metastatic CCRC patients to receive either sunitinib (50 mg/day 4 weeks on, 2 weeks off) or IFNα at a dose of 9 MU three times a week. The study primary endpoint was median PFS. Median PFS was significantly longer in patients assigned to the sunitinib treatment (11 months vs. 5 months) (HR 0.42 (95% CI, 0.32–0.54; *p* < 0.001)). The objective response in the sunitinib arm was 31%. A clear activity in terms of OS was not observed (26.4 vs. 21.8 months; HR 0.821; 95% CI, 0.673–1.001; *p* = 0.051), possibly due to the cross-over to the experimental arm of patients with progressive disease. The main side effects observed in the sunitinib group were hypertension, vomiting, diarrhea, hand–foot syndrome, and neutropenia [36]. 

Based on these results, sunitinib was FDA and European Medicines Agency (EMA) approved in 2006 for the first-line treatment of metastatic renal cell carcinoma. The standard dose is 50 mg/day, 4 weeks on, 2 weeks off. 

#### 3.2.2. Pazopanib

Pazopanib is an oral TKI active against VEGFR 1, 2, 3, c-kit, FGF receptor, and PDGFR hampering neo-angiogenesis and tumor growth [37]. 

In a phase III study enrolling 435 pre-treated or treatment-naïve metastatic CCRC patients, pazopanib demonstrated significant activity compared to the placebo. The results showed a PFS for treatment-naïve patients of 11.1 versus 2.8 months, and for cytokine-refractory patients of 7.4 versus 4.2 months, for pazopanib versus placebo, respectively. In addition, an ORR of 30% versus 3% was observed in the pazopanib arm compared with the placebo. No differences in OS were observed, but cross-over was allowed. Common pazopanib side effects included diarrhoea (52%), hypertension (40%), hair color changes (38%), and nausea (26%) [38]. The phase III, randomized, open-label trial COMPARZ study compared pazopanib and sunitinib as first-line agents. Both agents performed similarly. The non-inferiority endpoint of PFS was reached (8.4 vs. 9.5 months in the pazopanib and sunitinib arms, respectively; HR = 1.05 (95% CI, 0.90–1.22) and a similar OS was observed in all IMCD risk groups [39]. A similar PISCES study reported on patient-reported outcomes regarding pazopanib and sunitinib tolerability and patient preference. The study demonstrated equal activity, but different toxicity profiles (hepatotoxicity and diarrhoea for pazopanib; fatigue and hand-foot syndrome for sunitinib) and a better quality of life (QoL) for the pazopanib treatment [40].

As a second-line therapy after another TKI, pazopanib was studied in a phase II trial enrolling patients receiving first-line therapy with sunitinib or bevacizumab [41]. A total of 55 patients with metastatic CCRC were enrolled. The patients received 800 mg of pazopanib orally daily. Out of 55 patients (27%), 15 had an objective response to pazopanib. An additional 27 patients (49%) had stable disease, for a disease control rate of 76%. After a median follow-up of 16.7 months, the median PFS for the entire group was 7.5 months (95% confidence interval, 5.4–9.4 months), regardless of the previous treatment received. The estimated overall survival rate for the entire group at 24 months was 43%. 

EMA approved pazopanib as a first-line treatment for metastatic CCRC in adults.

#### 3.2.3. Sorafenib

Sorafenib is the second TKI approved for metastic CCRC. This drug is a multi-target TKI acting on VEGFR 1-3 PDGFR, c-kit, the serine-threonine kinase ros1, and stem cell receptors. More recently, other targets have been identified as CRAF, BRAF, V600E BRAF, c-kit, and FLT-3. The final result is a decrease in tumor angiogenesis and an inhibition of cell replication [42].

In a phase III TARGET trial, sorafenib (400 mg twice) was compared with a placebo in previously treated metastatic CCRC patients with at least one systemic treatment. The study was designed with OS as the primary endpoint. The final OS of the patients treated with sorafenib was not clearly superior compared with the OS of patients treated with the placebo (17.8 vs. 15.2 months, respectively; hazard ratio (HR) = 0.88; *p* = 0.146). The difference in terms of OS, however, became relevant in a further analysis censoring placebo patients with cross-over (17.8 vs. 14.3 months; HR = 0.78; *p* = 0.029) [43]. The ORR was 12% vs. 2% and the PFS was 9.8 months vs. 5.5 months in favor of sorafenib treatment. Hand-food syndrome, fatigue, and hypertension were the most relevant side effects of the drug [43]. An INTORSECT trial compared sorafenib with temsirolimus, an mTOR inhibitor, as a second line-therapy in metastatic CCRC. Sorafenib was superior in terms of OS (16.6 vs. 12.3 months), even if no significant differences were observed in the PFS (4.2 vs. 3.9 months) [44]. 

In a phase II study enrolling 189 untreated advanced CCRC patients, sorafenib did not improve the PFS when compared with IFN-α-2a [45].

The EMA approved sorafenib for use in the treatment of patients with advanced CCRC who have failed prior INFα or interleukin 2 therapy. The approved dose is 800 mg/daily in two refractory doses.

#### 3.2.4. Axitinib

Axitinib is a second-generation TKI active against VEGFR 1,2,3 [46]. All these receptors are involved in neo-angiogenesis, tumor growth, and metastatic progression. 

In two phase II studies, axitinib showed significant activity after INFα, interleukin 2, and sorafenib in metastatic CCRC as a second-line therapy [47,48]. In the first study of patients with cytokine-refractory metastatic CCRC, an ORR of 44.2% was achieved. The median time to progression was 15.7 months and the median OS was 29.9 months [47]. In the second study of sorafenib-refractory metastatic CRCC patients, the ORR was 22.6%. The median PFS was 7.4 months, and a median OS of 13.6 months was observed [48]. In a phase III AXIS trial [49], axitinib was compared to sorafenib in metastatic CCRC progressing to first-line sunitinib, cytokines, bevacizumab plus INF, or temsirolimus. The trial met its primary endpoint; axitinib was superior to sorafenib in terms of PFS (6.7 vs. 4.7 months), with no differences in survival (20.1 vs. 19.2 months). The reported side effects of axitinib were anemia, hypothyroidism, anorexia, headache, cough, proteinuria (21%), diarrhoea, hypertension, and fatigue [49].

The EMA approved axitinib at the dose of 10 mg/day for metastatic CCRC when sunitinib or cytokine treatment has failed. The recommended clinical starting dose for axitinib is 5 mg twice daily, taken with or without food. The dose increases up to a maximum of 10 mg twice daily, or a reduction is permitted based on individual tolerability [46].

#### 3.2.5. Cabozantinib

Cabozantinib is an oral TKI active against different receptors: mesenchymal epithelial transition factor (MET), VEGFR2, anexelekto (AXL), ROS proto-oncogene 1 (ROS1), protein tyrosine kinase 3 (TYRO3), MER, c-kit receptor, TRKB, FLT3, TIE-2, and RET. This agent acts on the microenvironment via MET, AXL, and VEGF inhibition, reducing motility, migration, invasion of tumor cells, and neo-angiogenesis [50]. The specific activity against VEGFR hampers angiogenesis in tumor tissue [50]. Considering these proprieties, cabozantinib cannot be considered as a “pure” antiangiogenic agent, but as a multitask agent interfering with cancer progression. 

In a large phase III METEOR study, cabozantinib was compared to everolimus in patients who had progressed on TKI. A significantly better PFS (7.4 vs. 3.8 months) and ORR (21% vs. 5%) was observed in the cabozantinib arm [51].

In the second study (CABOSUN), the drug was compared with sunitinib as a first-line treatment in poor and intermediate risk metastatic CCRC [52]. Previously untreated patients with advanced disease were randomized 1:1 to cabozantinib 60 mg daily or sunitinib 50 mg daily (4 weeks on/2 weeks off). The PFS was 8.2 months in the cabozantinib arm vs. 5 months in the sunitinib arm (adjusted hazard ratio, 0.66; 95% CI, 0.46–0.95; one-sided *p* = 0.012), and the ORR was 33% (95% CI, 23–44) for cabozantinib vs. 12% (95% CI, 5.4–21) for sunitinib. All grade 3 or 4 adverse events included diarrhoea (cabozantinib 10% vs. Sunitinib 11%), hypertension (28% vs. 22%), fatigue (6% vs. 15%), palmar–plantar erythrodysesthesia (8% vs. 4%), and hematologic toxicity (3% vs. 22%). The other main side effects are anemia, hypothyroidism, dysgeusia, headache, and dizziness. 

The EMA approved cabozantinib at the dose of 60 mg/day as a first-line treatment of patients with intermediate or poor risk metastatic CCRC, and as a second-line therapy for patients progressing to prior VEGF-targeted therapy.

### 3.3. mTOR Inhibitors

mTOR is a crucial component of the PI3K/AKT pathway, which modulates the angiogenesis process, cell proliferation, and metabolism either in normal or in cancer cells [53]. mTOR inhibition prevents the downstream of AKT and hampers HIF-1 expression and consequently neo-angiogenesis [53].

The known role of HIF1 in metastatic CCRC development and progression led to studies testing mTOR inhibitors in kidney carcinoma. Two mTOR inhibitors are approved in metastatic CCRC: temsirolimus and everolimus [54].

#### 3.3.1. Temsirolimus

Temsirolimus binds an intracellular protein, the peptidyl-prolyl cis-trans isomerase FKBP-2, inhibiting mTOR activity, and cellular replication [53].

A three arms pivotal study compared INFα vs. temsirolimus 25 mg alone vs. temsirolimus plus INFα in poor risk patients. Temsirolimus 25 mg once a week was superior in terms of ORR (8.6% vs. 4.8%) and PFS (5.5 vs. 3.1 months) when compared with INFα. Temsirolimus improved the median OS by 3.6 months. The combination arm did not improve the results. The side effects of temsirolimus included fatigue, viral infections, bacterial sepsis, cutaneous rush, mucositis, nausea, neutropenia, and renal toxicity [54].

In a second-line treatment study, 512 patients were randomly assigned 1:1 to receive intravenous temsirolimus 25 mg once weekly or oral sorafenib 800 mg daily. The stratification factors were: duration of prior sunitinib therapy, prognostic risk, histology (clear cell or non-clear cell), and nephrectomy status. The primary end point was PFS. The ORR and OS were secondary end points. Compared with sorafenib, temsirolimus showed a relative PFS benefit for patients with metastatic CCRC (4.6 vs. 3.9 months). Unfortunately, the median OS in the temsirolimus group was 12.2 months, and 16.6 months in the sorafenib group [55]. Accordingly, temsirolimus was not approved as a second-line treatment. The EMA approved temsirolimus (25 mg once a week) as a single agent for the first-line treatment of adult patients with poor risk metastatic CCRC who have at least a three out of six negative prognostic factor in accordance with the MSKCC classification.

#### 3.3.2. Everolimus

Everolimus is a selective mTOR inhibitor binding FKBP 12, an intercellular protein. This protein directly interferes with mTOR complex1, reducing its activity and signaling. mRNAs, which encode for proteins involved in glycolysis and in the cell cycle process, are consequently altered, and neoplastic cell proliferation is inhibited. In addition, everolimus reduces VEGF levels, inhibiting the angiogenetic process in the tumor [53]. 

In 2009, everolimus received EMA approval for use in the treatment of advanced CCRC patients progressing after first-line therapy on the basis of a RECORD I study. RECORD I was a phase III study comparing everolimus (10 mg daily) with a placebo in patients who progressed on TKI therapy (sorafenib, sunitinib, or both) [56]. The PFS was 4.9 months vs. 1.9 months and the ORR was 1.8% vs. 0% for everolimus and the placebo, respectively. In the everolimus arm, 63% of stable disease was reported. The most common side effects in the active arm were stomatitis, anemia, infections, neutropenia, cytopenia, headache, epistasis, proteinuria, interstitial pneumonitis, asthenia, abdominal pain, and rare, but severe, TKI pneumonitis [56].

Everolimus has been utilized as a comparator standard arm in many phase III studies on second-line therapy after TKI progression. For example, cabozantinib was approved after a METEOR study compared it with everolimus [51]. Similarly, nivolumab was approved showing a superior activity compared with everolimus in a second-line setting [8,57].

## 4. The Interplay between Angiogenesis and the Immune System: The Backbone for a New Era in CCRC Treatment 

The evidence in multiple studies demonstrates a strong interplay between angiogenesis and the immune system, both in physiological and pathological conditions. The inhibition of immune cell activity is crucial, for example, in the regulation of the physiological functions of VEGF in mediating the repair of wounds [58].

Metastatic CCRC is an immunogenic tumor with a high number of tumor-infiltrating lymphocytes (TIL) [8,59]. In addition, around 20–25% of renal cell cancers have a high expression of programmed cell death ligand 1 (PD-L1) [57,59]. Tumor immune evasion is widely influenced by angiogenesis. It is recognized that increased VEGF amounts in kidney cancer are able to induce the inhibition of the cells of innate and adaptive immune surveillance [60]. VEGF is known to interfere with dendritic cell maturation and the differentiation of progenitor cells into CD4^+^ and CD8^+^ T cells [61]. In addition, increased levels of VEGF induce an amplified number of myeloid-derived suppressor cells (MDSCs), cells with potent immune-suppressive activity against cytotoxic TIL [60,61]. The normalization of the number and architecture of tumor vessels favors immune cell infiltration, inducing major histocompatibility complex (MHC)-I antigen presentation, cytokine production, and a reduction in macrophages with inhibitory functions. In addition, the regulation of tumor vasculature can facilitate the uptake of other drugs, such as antibodies [62].

The rationale of the association of immunotherapy with an antiangiogenic agent consists in the modulation of immune microenvironments through enhanced T-cell priming and activation and the promotion of dendritic cells, and an increased number of tumor infiltrating lynphocytes (TIL) by blocking the tumor vasculature [56,57,58].

TKIs are known to interfere with the activity of immune cells [63]. Sunitinib has an immune-activating properties. It induces the inhibition of signal transducers and activates transcription 3 (STAT3) and c-kit100, causing a decrease in the number of regulatory T cells (Tregs) and MDSCs. Moreover, this drug favors the priming of T cells by dendritic cells and inhibits the production of co-inhibitory molecules, such as PD-1 and cytotoxic T-lymphocyte–associated antigen 4 (CTLA-4) [64]. Pazopanib and axitinib have a similar effect to sunitinib on the level of MDSCs. Treatment with axitinib in combination with dendritic cell (DC)-based vaccination promotes the activity of immune surveillance against tumor cells, activating tumor-specific CD8 T cells and decreasing MDSCs and Tregs [63,65]. MTOR inhibitors favor the differentiation and activity of Tregs [63]. Bevacizumab promotes the antigen-presenting activity of DCs, increasing T cell replication [63]. 

## 5. Immune Checkpoint Inhibitors in Combination with TKI and Immunotherapy

Recent studies have reported the impressive impact of the association between antiangiogenic agents and immunotherapy, and of immunotherapy combinations in metastatic CCRC. 

Six randomized controlled phase III trials investigated immune checkpoint inhibitors (ICI) in association with TKI or immunotherapy in advanced clear cell renal cancer, and many other studies are still ongoing [66,67,68,69,70,71,72]. In most of these studies, the comparator arm was sunitinib. We report the results of the published trials.

### 5.1. Nivolumab + Cabozantinib

Nivolumab is a PD-1 inhibitor humanized monoclonal antibody binding the programmed death 1 receptor and hampering its interaction with PDL1 and PD2. PD1 is a regulator of the T cell activity involved in immune response. Nivolumab increases T cell immune response, including T cell anti-tumoral responses [66,67]. 

In a phase III, randomized, open-label trial, patients with previously untreated clear cell advanced renal cell carcinoma were randomized to receive either nivolumab (240 mg every 2 weeks) plus cabozantinib (40 mg once daily), or sunitinib (50 mg once daily for 4 weeks of each 6-week cycle). The primary end point was PFS. The secondary end points included OS, objective response, and safety. At a median follow-up of 18.1 months, the median PFS was 16.6 months for the combination arm and 8.3 months for the sunitinib arm; the probability of OS at one year was 85.7% in the experimental treatment, and 75.6% in the sunitinib arm (HR for death 0.60; *p* < 0.001). The ORR was 55.7% in the patients receiving nivolumab plus cabozantinib vs. 27.1% in those treated with sunitinib. The activity of combination treatment was consistent across the subgroups, including IMDC risk stratification. Adverse events of any cause of grade 3 or higher occurred in 75.3% of the patients receiving the experimental treatment vs. 70.6% of those receiving sunitinib monotherapy [67]. The main side effects of the combination were infections, neutropenia, hypothyroidism, pancreatitis, hypersensitivity, and a loss of appetite [67].

In March 2021, the EMA approved the combination for the first-line treatment of advanced CRCC.

### 5.2. Avelumab + Axitinib

Avelumab is a fully human monoclonal antibody that inhibits the interplay between PD-1 on T cells and PD-L1 on tumor cells, reducing immunosuppression in the tumor microenvironment. The main side effects of the avelumab + axitinib combination are anemia, hypothyroidism, loss of appetite, headache, and peripheral neuropathy [66,68].

In a Javelin Renal 101 study, a total of 886 patients were randomized to receive avelumab (10 mg per kilogram intravenously every 2 weeks) plus axitinib (10 mg orally daily) or sunitinib (50 mg orally daily, 4 weeks on, 2 weeks off). The median PFS was 13.8 months for the combination arm compared with 8.4 months for the sunitinib arm. In PD-L1-positive tumors (63.2% of patients), the median PFS was 13.8 months with avelumab plus axitinib, compared with 7.2 months with sunitinib. In patients with PD-L1-positive tumors, the ORR was 55.2% with avelumab plus axitinib, and 25.5% with sunitinib. Adverse events of grade 3 occurred in 71.2% of patients treated with the combination, and in 71.5% of patients treated with sunitinib [68].

Based on these data, in 2019, the EMA approved the combination for the first-line treatment of metastatic CCRC.

### 5.3. Pembrolizumab + Axitinib

Pembrolizumab is a humanized monoclonal antibody anti PD-1 receptor, which hampers the interaction of the PD-1 receptor with PDL-1 and PDL-2. Pembrolizumab increases the anti-tumoral response of T cells [66,69]. 

In a randomized controlled phase III trial, KEYNOTE 426, 861 patients with untreated metastatic CCRC of all IMDC groups were assigned to receive pembrolizumab (200 mg intravenously once every 3 weeks) plus axitinib (10 mg/day) (432 patients) or sunitinib (50 mg orally for 4 weeks yes and 2 weeks not) (429 patients). The primary end points were OS and PFS. The secondary end point was ORR. After a median follow-up of 12.8 months, the ORR was 59.3% in the experimental arm and 35.7% in the standard arm [69]. The median PFS was 15.1 months for the combination group vs. 11.1 months for the sunitinib group; the hazard ratio for disease progression or death was 0.69 [69]. The activity of pembrolizumab plus axitinib in terms of OS and PFS was observed in all subgroups examined, including all IMCD risk groups (favorable, intermediate, and poor) and independently from PD-L1 expression. The side effects reported were as expected for the pembrolizumab plus axitinib combination: pulmonary infections, anemia, thrombocytopenia, anaphylaxis, hypothyroidism/hyperthyroidism, loss of appetite, hypocalcemia, peripheral neuropathy, cardiac arrhythmia, diarrhoea, epistasis, skin rash, myalgia, arthralgia, and fatigue [69].

The EMA approved the combination for the first-line treatment of advanced renal cell carcinoma in adults.

### 5.4. Atezolizumab + Bevacizumab

Atezolizumab is a humanized antibody against PDL-1. It does not act against PDL-2 and PD-1. The interaction with PDL-1 inhibits cell proliferation and cytokine production. The main side effects of the combination of atezolizumab and bevacizumab are: pulmonary infections, anemia, thrombocytopenia, neutropenia, leucopenia, lymphocytopenia, hypothyroidism, loss of appetite, peripheral neuropathy, headache, hypertension, cough, nausea, diarrhoea, liver enzyme elevation, cutaneous rash, proteinuria, and fatigue [70]. 

In a phase III, randomized IMmotion 151 study, 915 patients were assigned 1:1 to receive either atezolizumab (1200 mg) plus bevacizumab (15 mg/kg intravenously once every 3 weeks), or sunitinib (50 mg orally once daily for 4 weeks on, 2 weeks off). The median follow-up was 15 months. In the PD-L1 positive population, the median PFS was 11.2 months in the combination group vs. 7.7 months in the sunitinib group. The related grade 3–4 adverse events were 5% in the atezolizumab plus bevacizumab group and 8% in the sunitinib group [70].

### 5.5. Lenvatinib + Pembrolizumab or Everolimus

Lenvantinib is a selective TKI acting against different receptors: VEGFR1, VEGFR2, VEGFR3, FGFR 1,2,3 and 4, PDGFRα, c-kit, and RET. The main side effects of levantinib plus pembrolizumab are: urinary tract infections, thrombocytopenia, hypocalcemia, hypokalemia, dysgeusia, headache, hemorrhage, diarrhoea, nausea and vomiting, hand-foot syndrome, arthralgia, myalgia, fatigue, and peripheral edema [71].

In a phase III trial CLEAR study, in a 1:1:1 ratio, patients with untreated advanced renal cell carcinoma were randomized to receive lenvatinib (20 mg orally once daily) plus pembrolizumab (200 mg intravenously once every 3 weeks), lenvatinib (18 mg orally once daily) plus everolimus (5 mg orally once daily), or sunitinib (50 mg orally once daily, 4 weeks on and 2 weeks off). A total of 1069 patients were enrolled: 355 patients in the lenvatinib plus pembrolizumab arm, 357 patients in the lenvatinib plus everolimus arm, and 357 patients in the sunitinib arm. The study met its primary endpoint, with lenvatinib plus pembrolizumab significantly improving PFS compared with sunitinib (hazard ratio (HR) = 0.39, 95% confidence interval (CI) = 0.32–0.49; median = 23.9 vs. 9.2 months). Lenvatinib plus pembrolizumab also increased the ORR compared to sunitinib (71.0% vs. 36.1%) with an impressive complete response rate of 16.1%. The OS was significantly longer with lenvatinib plus pembrolizumab than sunitinib (HR = 0.66, 95% CI = 0.49–0.88). The lenvatinib plus everolimus arm significantly improved the PFS compared with sunitinib (median = 14.7 vs. 9.2 months, HR = 0.65, 95% CI = 0.53–0.83), but the OS benefit was inconclusive (HR = 1.15, 95% CI = 0.88–1.50). Grade 3 or higher adverse events were observed in 82.4% of the patients who received lenvatinib plus pembrolizumab, in 83.1% of those who received lenvatinib plus everolimus, and in 71.8% of those who received sunitinib. The most common grade 3 side effects included hypertension, diarrhoea, and increase of lipase levels [71].

### 5.6. Nivolumab + Ipilimumab

Nivolumab is a PD-1 immune checkpoint inhibitor antibody approved as monotherapy for the treatment of advanced renal cell carcinoma after treatment with antiangiogenic therapy, on the basis of an OS benefit [8].

Ipilimumab, an anti-CTLA4 antibody, is approved for the treatment of metastatic melanoma [1]. The inhibition of CTLA4 increases the number and the activity of T cells against tumoral cells and reduces the number of Treg cells. The combination of nivolumab + ipilimumab can cause the following main side effects: pain, anemia, lymphocytopenia, hypophysitis, loss of appetite, dehydration, hypokalemia, dizziness, blurred vision, diarrhea, nausea, vomiting, hypotension, gastrointestinal hemorrhage, hepatic toxicity, myalgia, arthralgia, and renal failure [72].

In a CheckMate 214, open label, phase III study, 1059 patients were 1:1 randomized to receive a combination of nivolumab and ipilimumab, or sunitinib [73]. At a median follow-up of 25.2 months in intermediate- and poor-risk patients, the 18-month OS rate was 75% with nivolumab plus ipilimumab, and 60% with sunitinib; the median OS was not reached with nivolumab plus ipilimumab versus 26.0 months with sunitinib [73]. The median PFS was 11.6 months and 8.4 months, respectively. Treatment-related adverse events occurred in 509 out of 547 patients (93%) in the combination group, and in 521 out of 535 patients (97%) in the sunitinib group; grade 3 or 4 events occurred in 250 patients (46%) and 335 patients (63%), respectively [73].

The EMA approved the combination of nivolumab plus ipilimumab in the first-line treatment of patients with intermediate and poor risk advanced renal cell carcinoma.

## 6. Discussion and Conclusions

Nowadays, the crucial role of angiogenesis in the development and metastatic spread of clear cell carcinoma is well known. As a consequence, antiangiogenic agents able to deeply interfere with several proangiogenic factors have changed the prognosis of this disease in the last 15 years, mainly in advanced or metastatic patients [1]. Controversial results have been reported in adjuvant settings [1]. TKIs are the most common agents used in daily practice in metastatic CCRC, and are the cornerstone of the new renal cancer therapeutic landscape, in association with immunomodulatory checkpoint inhibitors. Recently, innovative combinations of ICI or ICI with TKI have dramatically changed the first-line treatment landscape.

With the plethora of new therapeutic options, and considering the differences among the populations of the studies and the lack of mature OS data, clinicians face crucial challenges. The most important challenge is the correct sequence of treatment for each patient in order to personalize the therapy. At present, algorithms for treatment decisions rely on the IMDC risk model (Table 2) to stratify patients with untreated metastatic CCRC into different risk categories (Table 3 and Table 4) [1]. The large number of drugs now available for metastatic CCRC treatment can be proposed in the first and further lines of therapy. In Table 5, we recall all the above-mentioned drugs and the relative studies that led to their approval in clear cell carcinoma of the kidney. In second-line settings, the choice depends on the therapy previously given, and a second-line antiangiogenic TKI or checkpoint inhibitors can be alternatively considered. In case of progression after antiangiogenic drugs, the Checkmate 025 trial comparing nivolumab and everolimus showed an improvement in the median OS in the nivolumab arm [8]. After first-line ICI, any antiangiogenic drug is recommended by the European Association of Urology, but the level of evidence is very low [74,75]. Nowadays, because of the immaturity of the recent studies, definitive data are not available hampers to know which is the best therapeutic choice after the failure of the combination (Figure 1). For second-line therapies, we have many options, as represented in Figure 2. Beyond second-line treatment, enrolment into clinical trials is recommended whenever possible. However, based on recent trials with nivolumab and cabozantinib, different scenarios should be defined [76].

In conclusion, angiogenesis, and its regulatory mechanisms, is probably the most extensively studied target in oncology, and still represents an area of intensive research in renal cancer therapy. Since 2007, antiangiogenic drugs have deeply changed the treatment and the prognosis of patients with metastatic CCRC. Many questions are still unanswered and many areas remain to be explored, such as the research of predictive biomarkers, patient selection, how to overcome the resistance mechanisms, and the best sequence of therapies in daily clinical practice [75]. All these fields of investigation are a challenge to further improve the treatment efficacy in metastatic CCRC.

## Figures and Tables

**Figure 1 cancers-13-05896-f001:**
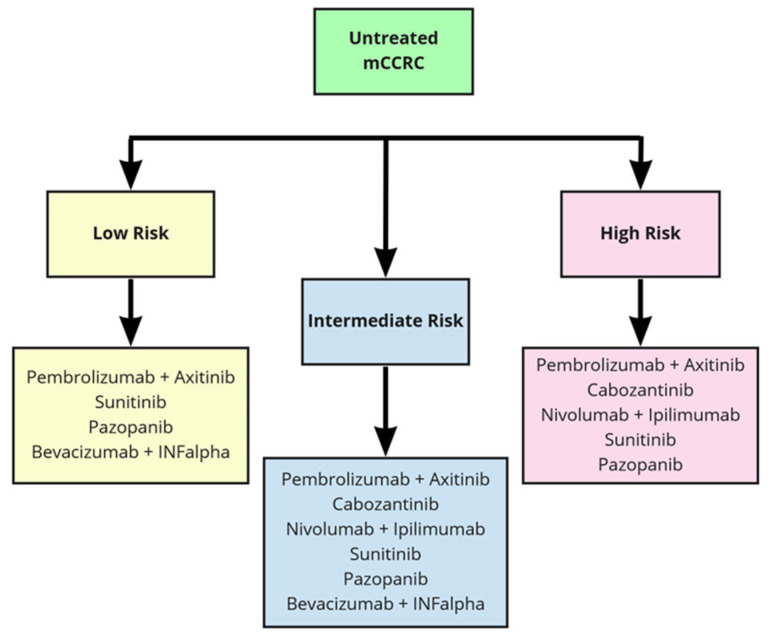
First line therapy in metastatic clear cells renal carcinoma (mCCRC).

**Figure 2 cancers-13-05896-f002:**
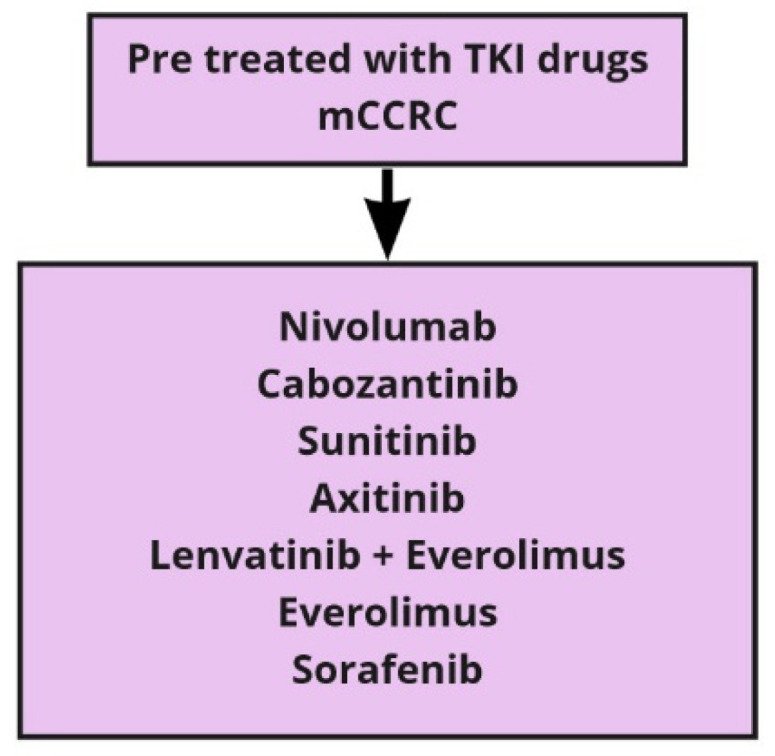
Second line therapy in metastatic clear cells renal carcinoma (mCCRC).

**Table 1 cancers-13-05896-t001:** Memorial Sloan–Kettering Cancer Center (MSKCC) renal cancer prognostic classification.

Factor	Poor Prognostic Factor
Time from diagnosis to treatment	<12 months
Hemoglobin	Lower limit of laboratory’s reference range
LDH	>1.5 the upper limit
Corrected serum calcium	10 mg/dL *
KPS	<80

* to express the pathological level of the sierum calcium is a necessary specification.

**Table 2 cancers-13-05896-t002:** International Metastatic RCC Database Consortium (IMDC) renal cancer prognostic score.

Risk Factors	Cut-Off Point Used
KPS	<80%
Time from diagnosis to treatment	<12 months
Haemoglobin	<lower limit of laboratory reference range
Corrected serum calcium	>10.0 mg/dL (2.4 mmol/L)
Neutrophilia	>upper limit of normal
Trhombocytosis	>upper limit of normal

**Table 3 cancers-13-05896-t003:** First-line treatment options in metastatic CCRC.

	Standard of Care	Alternative in Patients Who Cannot Receive Immunotherapy
IMDC favourable risk	Nivolumab/cabozantinib Pembrolizumab/axitinib Pembrolizumab/lenvatinib	Sunitinib Pazopanib
IMDC intermediate and poor risk	Nivolumab/cabozantinib Pembrolizumab/axitinib Pembrolizumab/lenvatinib Nivolumab/ipilimumab	Cabozantinib Sunitinb pazopanib

**Table 4 cancers-13-05896-t004:** Definition of prognostic criteria for International Metastatic RCC Databese Consortium (IMDC) and MSKK scores.

Prognosis	Score
Good	0
Intermediate	1–2
Poor	3–5

**Table 5 cancers-13-05896-t005:** Single drugs and combinations approved for metastatic CCRC.

Trial	Authors	Endpoint	mOSHR (95% IC)	mPFSHR(95% IC)	ORR
Sunitinib vs. INF alfa	Motzer et al., 2009 [36]	Primary: PFS; ORRSecondary: OS	26.4 ms vs. 21.8 HR 0.821; *p* = 0.051	11 vs. 5 months HR 0.42*p* < 0.001	31% vs. 3%
Pazopanib vs. IL2 or INF alfa	Sternberg et al., 2010 [38]	Primary: PFSSecondary:OS, ORR	No differences	7.4 vs. 4.2 monthsHR 0.46*p* < 0.0000001	30% vs. 3%
Cabozantinib vs. everolimus METEOR study	Choueiri et al., 2016 [51]	Primary: PFS Secondary: OS; ORR	No differences	7.4 vs. 3.8 monthsHR 0.58*p* < 0.0001	21% vs. 5%
Nivolumab plus cabozantinib vs. sunitinb	Choueiri et al., 2020 [68]	Primary:PFSSecondary:OS; ORR	Probability of OS at 12 months: 85.7% vs. 75.6%HR 0.60*p* = 0.001mOS not reached	16.6 vs. 8.3 monthsHR 0.51*p* < 0.001	55% vs. 27.1%
Pembrolizumab plus lenvatinib vs. lenvatinib plus everolimus or sunitinb	Motzer et al., 2021 [71]	Primary: PFSSecondary:OS; ORR	mOS not reached; HR for death (lenv + pem vs. suni): 0.66 *p* = 0.005	23.9 vs. 9.2 months (sunitinb) and 14.7 (everolimus)HR 0.39*p* < 0.001	71% vs. 53% vs. 36.1%
Pembrolizumab plus axitinib vs. sunitinibKeynote 426	Rini et al., 2019 [69]	Primary: OS; PFSSecondary: ORR; DOR; safaty	NR vs. 35.7HR 0.68(0.55–0.85)*p* < 0.001	15.4 vs. 11.1 monthsHR 0.71(0.60–0.84)*p* < 0.001	60% vs. 40%*p* < 0.0001
Nivolumab plus ipilimumab vs. sunitinb	Motzer et al., 2018 [73]	Primary: OS; PFS; ORRSecondary: ORR; OS PFS in ITT population; safety	NR vs. 26 monthsHR 0.63*p* < 0.001	11.6 vs. 8.4 months HR 0.82*p* = 0.03	42% vs. 27%
Bevacizumab + INF alfa vs. INFalfa + placeboAvoren study	Escudier et al., 2007 [23]	Primary: PFSSecondary: ORR; OS	No differences	10.2 vs. 5.4 months HR 0.63*p* = 0.0001	25% vs. 13%CALGB study
Sorafenib vs. placebo	Escudier et al., 2009 [45]	Primary: OSSecondary: PFS	17.8 vs. 14.3HR 0.78*p* = 0.029	5.5 vs. 2.8 months HR 0.44*p* < 0.00001	12% vs. 2%
Everolimus vs. placebo	Motzer et al., 2010 [56]	Primary: PFSSecondary: ORR; OS	No differences	4.9 vs. 1.9 months HR 0.55*p* < 0.0001	1.8% vs. 0%
Temsirolimus vs. INF alfaPoor prognosis pz	Hudes et al., 2007 [54]	Primary: PFSSecondary: ORR; OS	10.9 vs. 7.3 months HR 0.73*p* = 0.008	5.5 vs. 3.1 months HR 0.82*p* < 0.0001	8.6% vs. 4.8%
Axitinib vs. sunitinb	Rini et al., 2011 [49]	Primary: PFSSecondary: ORR; OS	No differences	6.7 vs. 4.7 months HR 0.0665*p* < 0.0001	8.6% vs. 4.8%
Atezolizumab + bevacizumab vs. sunitinibPhase II; ongoing phase III	Atkins et al., 2019 [70]	Primary: PFSSecondary: ORR; OS	Immature data	14.7 vs. 8.4 months 11.2 vs. 7.7 in PDL1 positiveHR 0.64, *p* = 0.0095

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
