# Peer review of "Antiangiogenic Therapy in Clear Cell Renal Carcinoma (CCRC): Pharmacological Basis and Clinical Results"

_cancers, 2021, doi:10.3390/cancers13235896_

Round 1

Reviewer 1 Report

In this review, Comandone et al. discuss recent advances in antiangiogenic therapy in clear cell Renal carcinoma. This article covers all relevant literature and drugs and molecular targets. However, I would recommend to also include a detailed discussion about adverse events in antiangiogenic therapy, particularly checkpoint inhibitors and TKIs. This should include organ complications especially the kidney itself as previously reported (Manohar et al. Kindey360 2020, Murakami et al. Kidney International 2021). Moreover, pathomechanisms including target expression should also be discussed (Cassol et al. KI Reports 2019, Hakroush et al. Frontiers Immunology 2020). 

Author Response

  • …I would recommend to also include a detailed discussion about adverse events in antiangiogenic therapy, particularly checkpoint inhibitors and TKIs. This should include organ complications especially the kidney itself as previously reported…

We thank the Reviewer for the suggestion. We added details about adverse events for each drug mentioned and in particularly we added adverse events for TKIs. Moreover, for the drugs with renal toxicity we pointed at  the main adverse events on renal function.  

  • …pathomechanisms including target expression should also be discussed…

We thank the Reviewer for the suggestion. We added pathomechanisms in every chapter correcting the part in which there weren’t properly described.

Reviewer 2 Report

This is a very interesting review that summarizes the current therapeutic strategies in CCRC. I believe this work deserves publication, after some minor improvements:

1. I miss a table that would synthetically collect all the data described in the text. It would certainly be a valuable summary for the reader.

2. The presented Figure 1 is very sloppy, so it is difficult to draw any observations on its basis. I like the very concept of this drawing, but the authors have to work on its readability. Besides, this drawing is probably cut off in the manuscript under review.

3. Authors should expand all abbreviations on their first use. 

4. Legends to the tables present in the manuscript would also be helpful. 

Author Response

  • I miss a table that would synthetically collect all the data described in the text. It would certainly be a valuable summary for the reader.

We thank the Reviewer for the suggestion. We added in the text table number 5 where all the studies described are reported.

  • The presented Figure 1 is very sloppy, so it is difficult to draw any observations on its basis. I like the very concept of this drawing, but the authors have to work on its readability. Besides, this drawing is probably cut off in the manuscript under review.

We thank the Reviewer for the suggestion. We changed Figure 1 previously drew and we changed in two figures (Figure 1 and Figure 2) that can clearly represent the First and the second line therapy in metastatic clear cells renal carcinoma (mCCRC).

  • Authors should expand all abbreviations on their first use.

We thank the Reviewer for the suggestion. We expand all the abbreviation previously missed.

  • Legends to the tables present in the manuscript would also be helpful.

We thank the Reviewer for the suggestion. At the end of the paper we draw a table, called LEGEND OF TABLES AND FIGURES,  which is a synthetic resume of the tables and figures presented in the text.

Reviewer 3 Report

The authors thoroughly describe the available therapeutic options and standard of care for renal cell carcinoma. The content of the review is complete and of high interest, with few articles doing such a quality review of this topic. However, I believe the article is a little hard to read as it stands. My recommendations for the authors:

Please, ensure your paragraph structure encompasses the topics you are describing. (i.e., 1 main point, 1 paragraph) As of now, many sentences seem unconnected to the text, distracting the reader. Your average subsection should present 1-3 paragraphs at most, and your transitions should accompany the reader toward the main point of the section.

I found the "mTOR inhibitor", "angiogenesis: the key player in development and progression of kidney cancer" and "small molecules and tyrosine kinase inhibitors" to be especially disorganized and hard to read. I believe merely reorganizing the contents for readability should suffice to improve the value of this review.

Finally, creating a table with the name of the study, dosages, frequency and timeline of clinical trials should help clarify the sections reviewing them.

A few more minor points, in no particular order:

1) Please, check that all your acronyms are defined.

2) Please, cite the clinical study only, no need to specify the authors of the publication.

3) Please, check your formatting. (e.g., line 260).

4) I would edit line 85 to read "angiogenesis is a physiologic process required to generate new blood vessels". The additional words in this sentence are technically inaccurate and do not add much to a definition that should, in my opinion, be concise.

5) I detected a lot of reference redundancies in the text. Increasing the textual coherence (i.e., fewer paragraphs and a few more connectors to help make a point better) would help remove references adding several times in consecutive sentences.

6) in line 25, the authors make a point that RCC tumours are "extremely angiogenesis-dependent". However, I think this statement should be toned down or discussed with more nuance according to 10.1016/j.ccell.2020.10.021.

7) in line 326, y recommend that authors begin with "for example" their sentence relating to cabozantinib approval. Otherwise, the sentence looks out of place.

8) could the authors discuss what they meant in sentence 462?

9) A few sentences had inaccurate uses of English words: e.g., line 93. I also found a few typos (e.g., line 49) and a few seemingly unconnected sentences (e.g., line 46). Please, revise your spelling, use of English and readability.

Author Response

MAJOR POINTS

1) Please, ensure your paragraph structure encompasses the topics you are describing. (i.e., 1 main point, 1 paragraph) As of now, many sentences seem unconnected to the text, distracting the reader. Your average subsection should present 1-3 paragraphs at most, and your transitions should accompany the reader toward the main point of the section.

We thank the Reviewer for the suggestion. We deeply modified the structure of each subsection as suggest: one main point, one paragraph. We reduced the paragraphs number of each subsection.

2) I found the "mTOR inhibitor", "angiogenesis: the key player in development and progression of kidney cancer" and "small molecules and tyrosine kinase inhibitors" to be especially disorganized and hard to read. I believe merely reorganizing the contents for readability should suffice to improve the value of this review.

We thank the Reviewer for the suggestion. We deeply modified the structure of the cited subsections.

3) Finally, creating a table with the name of the study, dosages, frequency and timeline of clinical trials should help clarify the sections reviewing them.

We thank the Reviewer for the suggestion. We made a table with the required characteristics.

MINOR POINTS

1) Please, check that all your acronyms are defined.

We thank the Reviewer for the suggestion. We redefined all acronyms.

2) Please, cite the clinical study only, no need to specify the authors of the publication.

We thank the Reviewer for the suggestion. We deleted the name authors in the text.

3) Please, check your formatting. (e.g., line 260).

We thank the Reviewer for the suggestion. We redefined the formatting text.

4) I would edit line 85 to read "angiogenesis is a physiologic process required to generate new blood vessels". The additional words in this sentence are technically inaccurate and do not add much to a definition that should, in my opinion, be concise.

We thank the Reviewer for the suggestion. We modified the sentence as requested.

5) I detected a lot of reference redundancies in the text. Increasing the textual coherence (i.e., fewer paragraphs and a few more connectors to help make a point better) would help remove references adding several times in consecutive sentences.

We thank the Reviewer for the suggestion. We removed several references.

6) in line 25, the authors make a point that RCC tumours are "extremely angiogenesis-dependent". However, I think this statement should be toned down or discussed with more nuance according to 10.1016/j.ccell.2020.10.021.

We thank the Reviewer for the suggestion. We discussed this point with more nuance as suggested.

7) in line 326, y recommend that authors begin with "for example" their sentence relating to cabozantinib approval. Otherwise, the sentence looks out of place.

We thank the Reviewer for the suggestion. We modified the sentence as suggested.

8) could the authors discuss what they meant in sentence 462?

We thank the Reviewer for the suggestion. We modified the sentence.

9) A few sentences had inaccurate uses of English words: e.g., line 93. I also found a few typos (e.g., line 49) and a few seemingly unconnected sentences (e.g., line 46). Please, revise your spelling, use of English and readability.

We thank the Reviewer for the suggestion. We revised English language in the text. In addition we have corrected some typos.  

Round 2

Reviewer 1 Report

The authors addressed all my comments, I recommend acceptance.